# Environmentally friendly polymers are used to enhance the water retention capacity of waste residue and the potential for vegetation growth

Jun Kang Zhao[1,2], Qun Chen[1,2]*, Lu Li[3], Cheng Zhou[1,2], Ting Quan He[4]

1 State Key Laboratory of Hydraulics and Mountain River Engineering, College of Water Resource and Hydropower, Sichuan University, Chengdu, Sichuan, China, 2 College of Water Resource and Hydropower, Sichuan University, Chengdu, Sichuan, China, 3 PowerChina Chengdu Engineering Corporation Limited, Chengdu, Sichuan, China, 4 Guangxi New Development Transportation Group Co., Ltd., Nanning, Guangxi, China

* chenqun@scu.edu.cn

## Abstract

The large pores and lack of water storage capacity limit the ecological rehabilitation of the waste residue. Modified hydrophilic polyurethane (W-OH) was used to improve the water retention of the residue. The water retention capacity of residues with different mass concentrations (1%, 3%, and 5%) of W-OH solution based on water mass was evaluated for several drying-wetting (D-W) cycles at 30°C and 50°C. The plant growth experiment investigated the plant growth status of waste residue before and after adding W-OH, to demonstrate the excellent water retention of W-OH under the same waste residue. Infiltration tests showed that W-OH effectively increased the volumetric water content (VWC) of waste residue and reduced its drying time. Untreated residue had 11.62% saturated initial VWC and 152 h drying time; 3% and 5% W-OH groups showed similar results (17.63%−18.09% VWC, 288–304 h drying time), so 3% W-OH is recommended for less usage. After 4 dry-wet cycles, both groups had reduced VWC and drying time. However, the 3% W-OH-treated residue still exhibited better water retention. A dense membrane is formed by W-OH solution around waste residue particles, enclosing/connecting particles, filling pores (reducing large ones), increasing capillary water storage, and hindering water evaporation. Degraded by more D-W cycles and higher temperature, the membrane weakens water retention, yet W-OH-treated residue still has better water retention than the untreated residue. Water retention of improved waste residue was enhanced, and plant growth promoted, per plant growth tests. After 21-day planting, Amorpha fruticosa in 3% W-OH-treated residue had better growth parameters than in untreated residue. This means that the waste grade treated with 3% W-OH can provide more water for plants to meet their growth needs.

**Data availability statement:** All relevant data are within the manuscript and its Supporting Information files.

**Funding:** This research was substantially supported by the Sichuan Province National Natural Science Foundation of China (Grant No. 24NSFSC0260); and the 3rd series of the Guangxi Transportation Keynote Research Program in 2020 (Grant No. XFZ-KY-LYH-20200112). The funder had no role in study design, data collection and analysis, decision to publish, or preparation of the manuscript. The specific roles of this author are articulated in the `author contributions' section.

**Competing interests:** We declare that we have no financial and personal relationships with other people or organizations that can inappropriately influence our work, and there is no professional or other personal interest of any nature or kind in any product, service and/or company that could be construed as influencing the position presented in, or the review of, the manuscript entitled. Sincerely yours, Jun Kang Zhao, Qun Chen, Lu Li, Cheng Zhou, Ting Quan He.

## Introduction

In the construction process of mining engineering and road construction, a large amount of waste residue with diverse shapes and sizes would be produced. The large blocks of rock are called rubble [1], waste rock [2–4], or crushed rock [5]. There are smaller fragments or even powders that may be produced during the process of blasting and excavation, such as crushed rock sand [6], sandstone cutting waste [7], and rock dust [8]. The generation and disposal of these solid wastes hurt the surrounding environment, which is difficult to recover, including dust pollution, noise pollution, soil pollution, and so on [8].

Some waste materials were reused in many engineering projects because of their good physical and mechanical properties. These new applications of solid waste not only reduce the discharge and accumulation of them but also make them into new and environmentally friendly building materials with diverse functions [3,4,7,9,10]. Large pieces of hard rock can be crushed and used as building materials and aggregates [6,7,9,10]. The hard Limestone and Basalt were crushed and used as railway ballast [9]. Smaller particles of crushed stone sand could be used as high-quality fine aggregate to replace river sand, and recycled stone powder could be used as raw material for ultra-high-performance concrete [6,11].

The above applications were all aimed at reusable waste rock blocks. However, most of the mining and rock waste cannot be reused under current production conditions due to its poor physical and mechanical properties. They were often piled up near mountains in abandoned soil and rock sites. Rock blocks with smaller particles were piled up on slopes on both sides of roads for vegetation restoration in arid areas due to a lack of soil [4]. Due to the shortage of water resources in arid areas and the evaporation of water caused by high temperatures, the water provided for plant growth is insufficient, and the soil nutrient content is low, it is difficult for plants to grow normally, and the vegetation restoration is limited [12]. Especially in some severe arid areas, rocky soil surfaces with poor water retention are prone to land degradation, desertification, and even non-uniform settlement due to long-term water scarcity [13].

At present, the methods to improve the water retention of soil mainly include physical, chemical, and biological methods. Biochar was the most commonly used water-retaining agent in physical water retention methods. It was mostly used to improve the water retention and erosion resistance of soils with smaller particles such as clay and soil [14–16]. Even the soft sock was utilized to improve the hydraulic characteristics of the sandy soil [17]. The Microbial induced carbonate precipitation (MICP), microbial extracellular polymeric substances (EPS), and bacterial polyextremotolerant bio-emulsifiers from the microorganisms were applied to improve water retention and crack resistance of fine particles such as sand and clay [18–20]. Chemical polymer materials, such as superabsorbent polymer (SAP), superabsorbent composites (SACs), polyacrylate polymer (PP), and polyacrylamide (PAM) were utilized to improve the water retention of soil [21–24]. Steel slag and biopolymer cellulose were used to improve the sustainable stabilization of sandy soils [25]. It can be found that the above water retention methods are all aimed at small particles of clay, fine

sand, and so on. For larger particles of residue, the study of the above materials is relatively limited. limited by the viscosity of these water-retaining materials, the larger particles cannot agglomerate, and there are still large pores among the particles, so water is easy to evaporate and it is difficult to maintain capillary water. Therefore, the most important issue in improving the water retention of sand and residue is to reduce the pores among the particles.

W-OH is widely used for soil solidification and protection due to its good viscosity, swelling, and water absorption properties [26,27]. It is incorporated into different types of soil, such as Pisha sandstone [28,29] and red clay slope [30]. This helps prevent soil erosion [31], reduces evaporation of the water [30], and suppresses soil cracking [32]. Through the chemical analysis, the W-OH gel can degrade into nitrogen, carbon dioxide, and water [33]. The toxicity tests show that the W-OH has no harmful effects on animals and plants [32], so it is a friendly chemical material for the ecological environment.

It is worth noting that chemical materials, especially synthetic polymer materials, are susceptible to destructive effects such as D-W cycles and freeze-thaw cycles, their structure is degraded, and their properties are significantly weakened [34,35]. Therefore, it is significant to study the effect of D-W cycles and temperatures on the performance of polymer water-retaining agents, for the application of water-retaining agents in arid and high-temperature areas, but there is still relatively little research on this topic at present.

W-OH has the potential to be useful in projects related to soil and water conservation and ecological restoration of waste residue soil, but there is little research on the influence of temperature and D-W cycles on the water retention of W-OH materials. To study the water retention and whether or not deteriorates when it is sustained with the drying-wetting (D-W) cycle, a laboratory experiment was conducted to study the effects of different concentrations of W-OH solution on the water retention characteristics of residue under the D-W cycles and different environmental temperatures, and the improvement mechanism of water retention of rock by adding W-OH was investigated. The plant experiments explored the potential of adding W-OH for the ecological restoration of waste residue. Overall, the study findings explored the effectiveness of the W-OH gel in vegetation restoration projects involving waste residue soil, filling the gap in improving the water retention of the waste residue.

## Materials and methods

### Materials

**Waste residue.** The experimental samples were taken from a residue disposal site (in Sichuan Province, China). The grading curve of residue particle size distribution was obtained through a screening test (Fig 1) according to the Standard for Geotechnical Testing Method [36]. The uniformity coefficient of the residue was 17.65, and its coefficient of curvature was 1.96. Table 1 shows the grading and physical property parameters of the residue. Its natural dry density was 1.75 g/cm$^3$. The coefficient of permeability obtained by the constant head permeability test was $5.2 \times 10^{-3}$ cm/s.

**W-OH.** Modified hydrophilic polyurethane resin (W-OH) is a light yellow, oily, viscous liquid produced by Jie Chengkai New Materials Co., LTD. (Jiangsu Province, China). With a density of 1.18 g/cm$^3$ [28] and the pH value is 6–7 [33], it can react with water in any proportion to form a solution and solidify in a short time to form a milky white porous elastic gel with good mechanical properties (Fig 2) [30].

Fig 3 illustrates how W-OH solidifies when it reacts with water. The reaction between W-OH molecules and water leads to the formation of a network polymer made up of units of W-OH (marked with a grey shadow in Fig 3a [37]). These simple network structure polymers then stack together and bond to create a complex porous elastic network structure (Fig 3c).

### Test scheme

Considering that W-OH and water react easily and quickly gel, the setting time and effect of 1%, 3% and 5% W-OH were compared in order to be close to the actual construction of drip irrigation. The result shows that the times for 1%, 3% and 5% W-OH forming gel were 252s, 135s, and 84s, respectively. When the W-OH concentration is below 1%, the solution

   

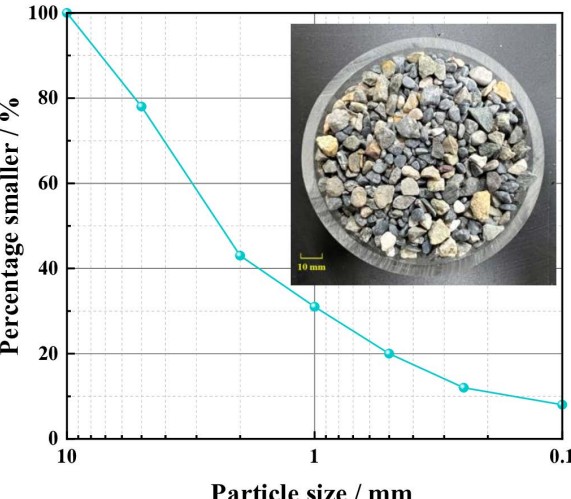

**Fig 1. Particle distribution curve of the residue.**

**Table 1. The physical property parameters of residue.**

| Parameter | $C_u$ | $C_c$ | $\rho_{max}$ (g/cm³) | $G_s$ | $k$ (cm/s) |
|-----------|-------|-------|----------------------|-------|------------|
| Value | 17.65 | 1.96 | 1.85 | 2.69 | $5.2 \times 10^{-3}$ |

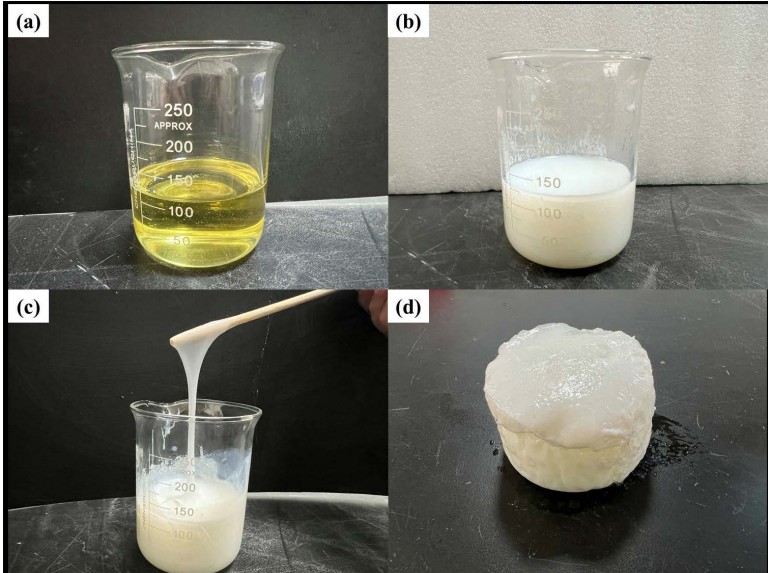

**Fig 2. Different states of W-OH material.** (a) Original W-OH (b) W-OH solution (40 seconds) (c) The state of W-OH gel (90 seconds) (d) W-OH gel (180 seconds).

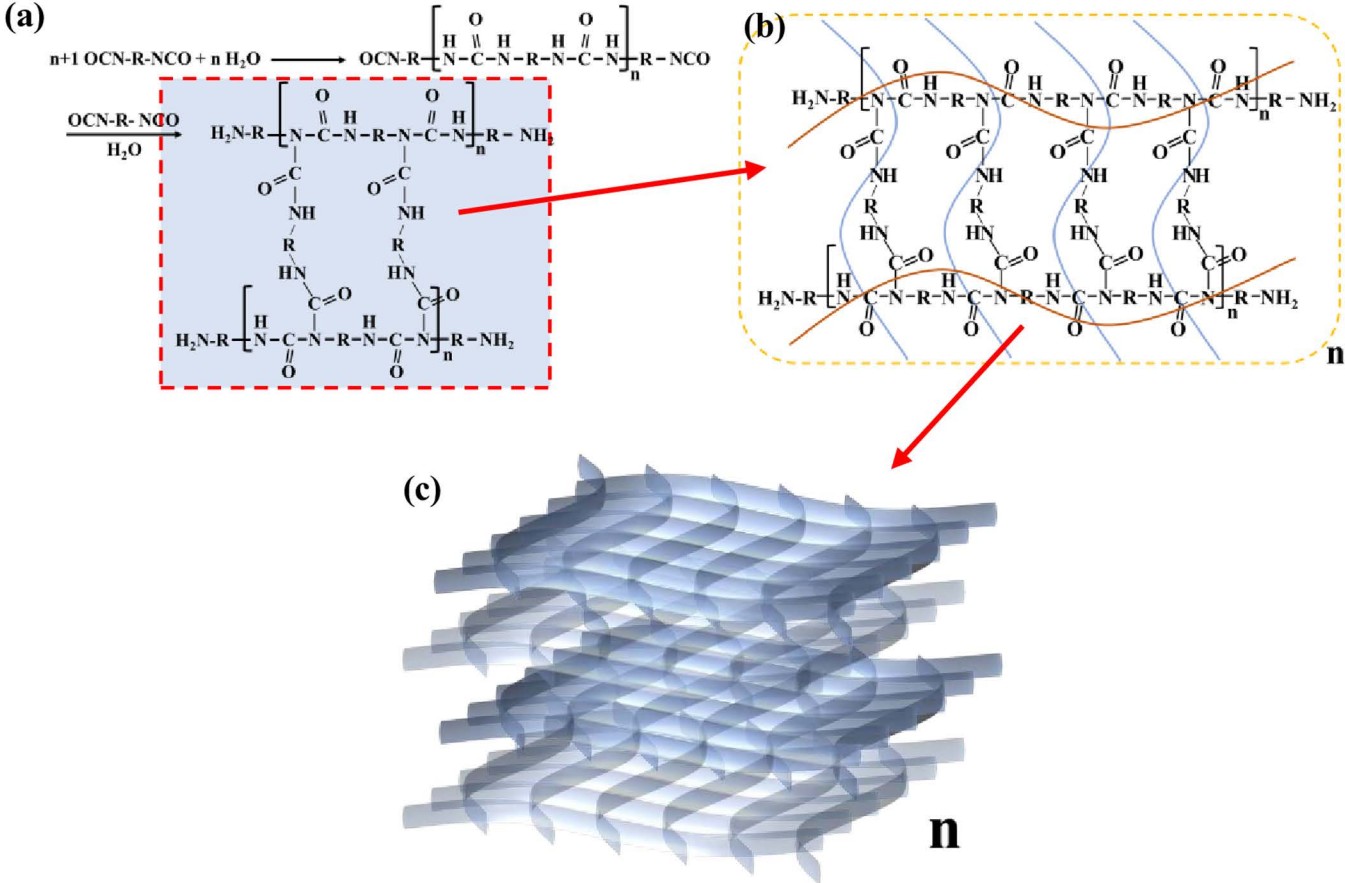

**Fig 3. Solidification process of W-OH with water.** (a) Reaction formula for W-OH solidification [37] (b) The network structure of W-OH gel (c) Elastic porous structure of W-OH gel.

exhibits low viscosity, resulting in ineffective bonding of larger sand and residue particles. At a 5% concentration, the W-OH solution is characterized by high viscosity and rapid gelation kinetics. However, when the concentration exceeds 5%, the infiltration capacity of the solution is significantly reduced, leading to hindered uniform dispersion and incomplete pore infiltration within the particle matrix. The shorter gel time is not conducive to the spraying or drip irrigation of the W-OH solution, which may lead to clogging of the nozzle of construction equipment and waste of W-OH materials. Therefore, the concentration of W-OH solution was set to 1%, 3%, and 5% in the experiment. Fig 4 shows the gelling state of 1%, 3% and 5% W-OH.

To verify the water retention of W-OH in extremely high temperature environments, the water retention effect of the residue before and after W-OH modification was studied under a temperature environment of 50°C and 4 W-D cycles. To verify the improvement effect of W-OH on the water retention of rubble, a plant growth experiment was conducted. The test program is shown in Table 2. The sample without W-OH gel was taken as a comparison case in all test groups, the added amount of the W-OH solution was 2.0 L/m² in all tests.

## Sample preparation and saturation

Fig 5 shows the preparation process of the samples. A self-made acrylic container with a diameter of 100 mm and a height of 50 mm with a perforated plate at the bottom was used to hold the residue particles. A filter paper was laid on the bottom

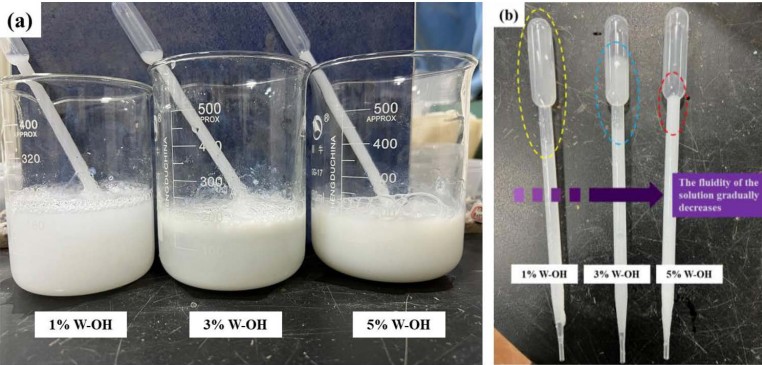

**Fig 4. Gelling state of 3% and 5% W-OH.**

**Table 2. Test scheme.**

| Test Group | Concentration of W-OH solution (%) | D-W cycles | Environmental temperature (°C) |
|---|---|---|---|
| 1 | 1, 3, 5 | 1 | 30 |
| 2 | 1, 3, 5 | 1, 2, 3, 4 | 30 |
| 3 | 1, 3, 5 | 1, 4 | 30, 50 |

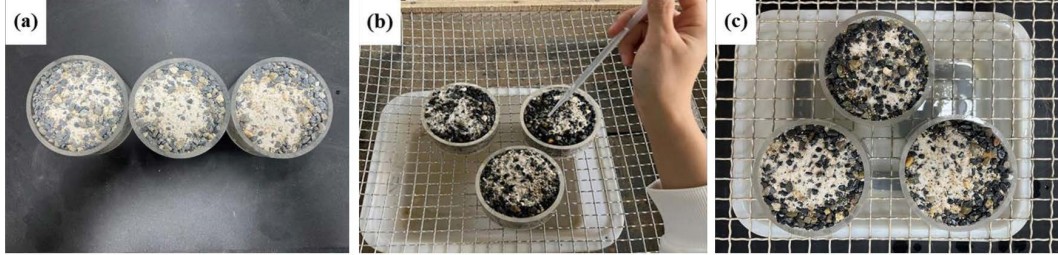

**Fig 5. Preparation of the residue samples.** (a) Prepared samples (b) Dripping water to saturation (c) Removing gravity water.

of the container, according to the measured natural dry density of the residue ($1.75\,g/cm^3$), and the residue was evenly filled into the container. The W-OH solution ($2.0\,L/m^2$) was dripped onto the residue surface. The W-OH solution penetrated the pores of the particles, and the stable W-OH gel was formed after 3 minutes. The surface of the sample is evenly dripped with deionized water to enable permeation into its interior. The dripping rate and volume are precisely controlled, and the amount of deionized water dripped into the sample is equal to the amount of water seeped out, which means the sample attained the maximum water content under natural drip irrigation conditions.

## Test methods

**Dry-wet cycle test at different temperatures.** To investigate the water retention durability of the W-OH gel, and the effect of environmental temperature. The saturated residue samples containing W-OH were placed in a drying oven at 30°C and 50°C, and the mass of the samples was weighed every 2 hours to calculate the volumetric water content of the samples. Once the volumetric water content of the samples dropped to zero, which means one of the D-W cycles was over. Deionized water is reapplied to the sample until the amount of water seeping out from the bottom of the sample is

equal to the amount of water dripping in, at which point the sample reaches saturation again. Then the subsequent wet-dry cycle test is initiated. This wet-dry cycling procedure is repeated four times to evaluate the water retention durability of W-OH-treated sand-residue composites. In the drying duration, the slope of the tangent between the initial volumetric water content of the sample and the drying time is used to define the drying rate | kdr | (%/ h) of the sample after different concentrations of improvement (eq. 1).

$$\left| k_{dr} \right| = \frac{volumetric\ water\ content\,(\%)}{drying\ duration\,(h)}$$

(1)

### Plant growth test

A plant growth test in the laboratory was conducted after the best concentration of the W-OH solution was determined. Amorpha fruticosa was chosen as the herbaceous plant for the experiment. Amorpha fruticosa grass seeds were sown in transparent plastic sample boxes with perforations at the bottom. The shape of the sample box is trapezoidal, with a height of 600 mm. The top length and width of the sample box are 170 mm and 117 mm respectively, the bottom length and width are 135 mm and 85 mm respectively, and the length and width of the box at a height of 40 mm are 156 mm and 106 mm respectively. Based on the density of the waste slag, the required mass to fill 40 mm is calculated, and the waste slag is loaded into the sample box. Evenly sprinkle 100 full particle Amorpha fruticosa grass seeds into the sample box containing waste residue, and then drip 2L/m$^2$ of water and 3% concentration of W-OH solution into the sample box.

The growth of plants in waste residue at a concentration of 3% and without W-OH for 21 days was studied. During the growth process of the plant, drip 2L/m2 of water into the sample box every 7 days. The condition of the plant roots was recorded, and the mass of both plant roots and stems was weighed using an analytical balance. The root top ratio [r/t] (Root biomass (dry weight)/ Aboveground biomass (dry weight)) was then calculated. It is worth noting that in the plant growth experiment, 10 plants were selected for the determination of growth parameters, and the average value was taken as the basis for judging the growth effect of plants in the waste Residue.

## Results and analysis

### Effect of W-OH concentration on the water retention of the waste residue

Fig 6 shows the volumetric water content and the drying rate ($|k_{dr}|$) of the samples with different concentrations of W-OH solution for the first D-W cycle. The history of volumetric water content and the drying duration of the samples increased with the increase in concentration of the W-OH solution. The initial volumetric water content of the samples with W-OH gel is over 16%, and the drying duration of the samples with W-OH gel is over 260 hours. The volumetric water content of the sample shows a non-linear decreasing trend throughout the drying process. The average drying rates of the samples with different concentrations of W-OH solution were also different. The drying rate ($|k_{dr}|$) of the samples with 0, 1%, 3%, and 5% W-OH solution is 0.0764, 0.0630, 0.0612, and 0.0526, respectively. The higher the concentration of the W-OH solution, the lower the drying rate of the samples.

Fig 7 shows the initial volumetric water content and the drying duration of the samples with different concentrations of W-OH solution for the first D-W cycle. The samples without W-OH gel have less initial volumetric water content and drying duration than those of the samples treated with W-OH gel. The initial volumetric water content of the D-W cycle is 11.6% for the sample without W-OH gel, while it is 16.37%, 17.6%, and 18.1% for samples with 1%, 3%, and 5% W-OH solution, respectively. It is 1.41, 1.50, and 1.56 times the initial volumetric water content of the sample without W-OH gel. The drying duration for samples without W-OH is 152 hours, whereas for samples with 1%, 3%, and 5% W-OH solution, it is 260 hours, 288 hours, and 304 hours, respectively, which is 1.71, 1.89, and 2.00 times that of the sample without W-OH gel.

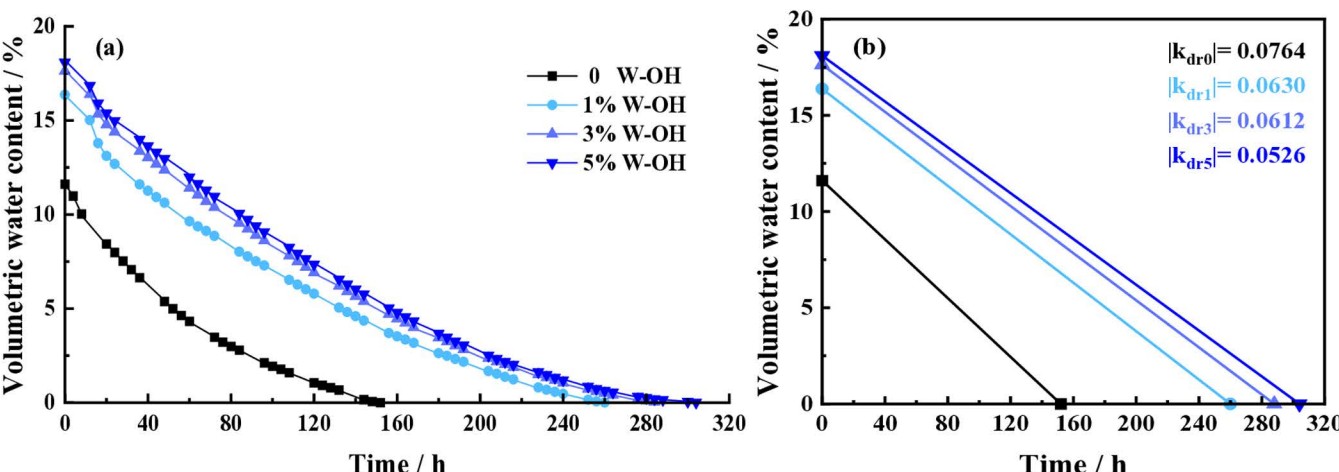

**Fig 6. The history of volumetric water content and the drying rate (|$k_{dr}$|) of the samples for the first D-W cycle.**

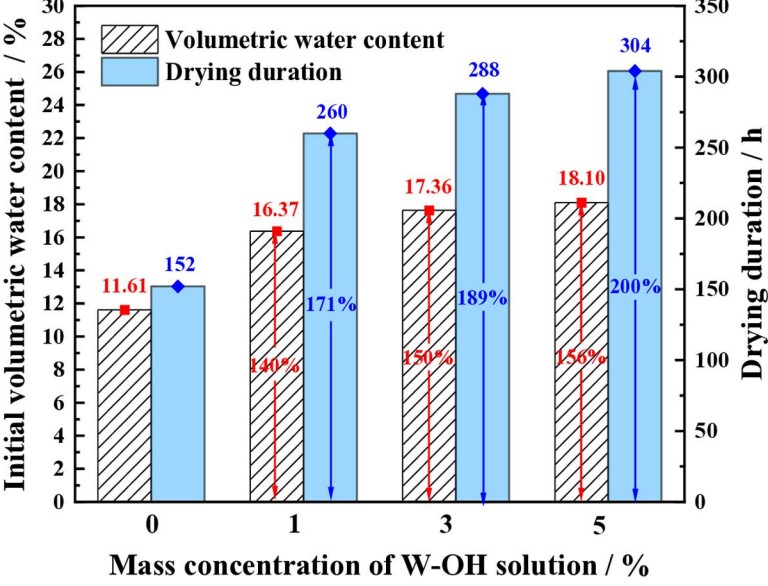

**Fig 7. The water retention of the samples in the first D-W cycle.**

## Effect of D-W cycles on the water retention of the waste residue

Fig 8 shows the initial volumetric water content and the drying duration of the samples for different D-W cycles. The initial volumetric water content of samples without W-OH gel for the first D-W cycle is 11.6. After the second D-W cycle, the initial volumetric water content of samples without W-OH gel is 9.6%, 9.2%, and 9.0%, respectively. The drying duration of the samples without the W-OH gel is 152 hours during the first D-W cycle. For the 2nd to 4th D-W cycle, the drying duration of samples without W-OH gel is 128 hours, 126 hours, and 124 hours, respectively. The water retention of the samples deceased with the increase in the D-W cycles. That means the system formed by the W-OH gel and the residue particles gradually deteriorates.

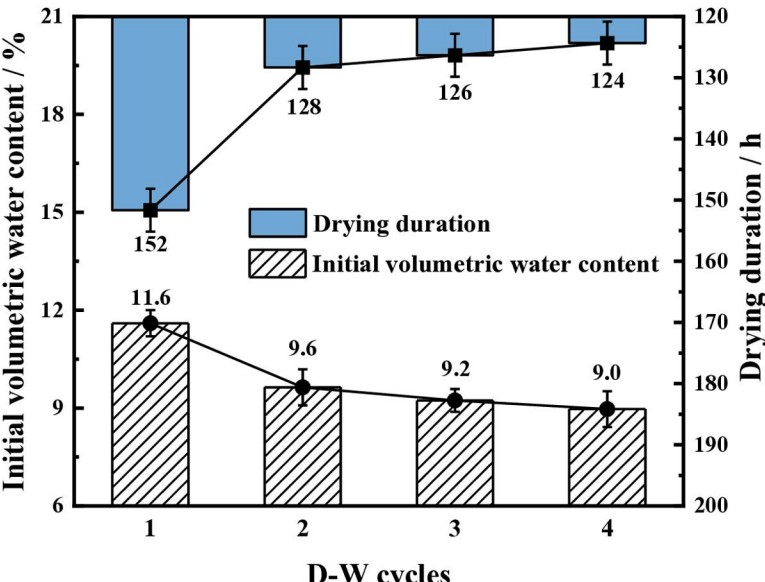

**Fig 8. The initial volumetric water content and drying duration of the samples for different D-W cycles.**

Fig 9 shows the history of volumetric water content of the samples, adding the W-OH solution of different concentrations for each D-W cycle. The volumetric water content of samples with different concentrations of W-OH solution gradually decreased during the 4 D-W cycles. The initial volumetric water content of the samples without W-OH is 11.62% for the first D-W cycle. The total drying duration of the samples without W-OH gel is 568 hours. The initial volumetric water content of samples with W-OH gel is 16.37% to 18.09% for the first D-W cycle, and the total drying duration is 1004 hours to 1148 hours.

Fig 10 shows the initial volumetric water content and drying duration of D-W cycles for the samples with different concentrations of W-OH solution at 30°C. The initial volumetric water content of the samples gradually increased with

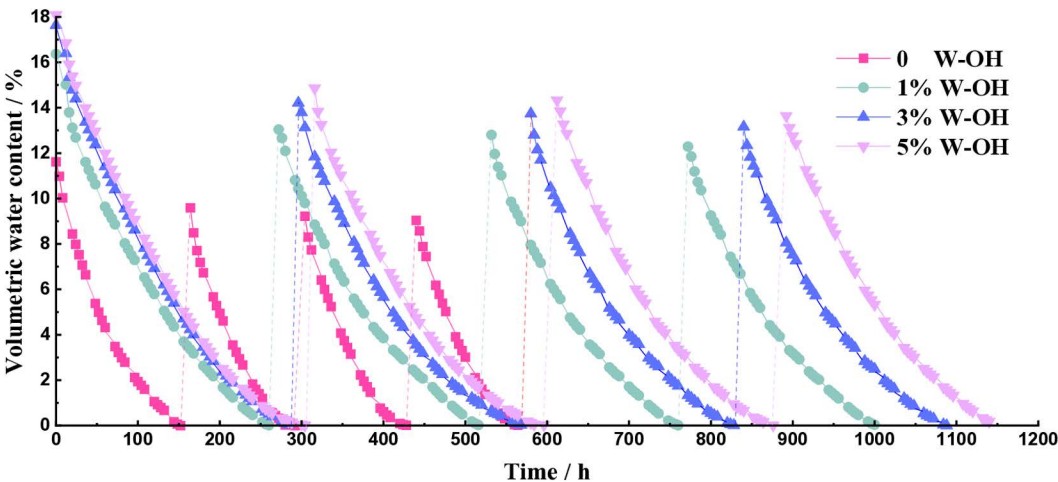

**Fig 9. The history volumetric water content of the samples added W-OH solution of different concentrations at each D-W cycle.**

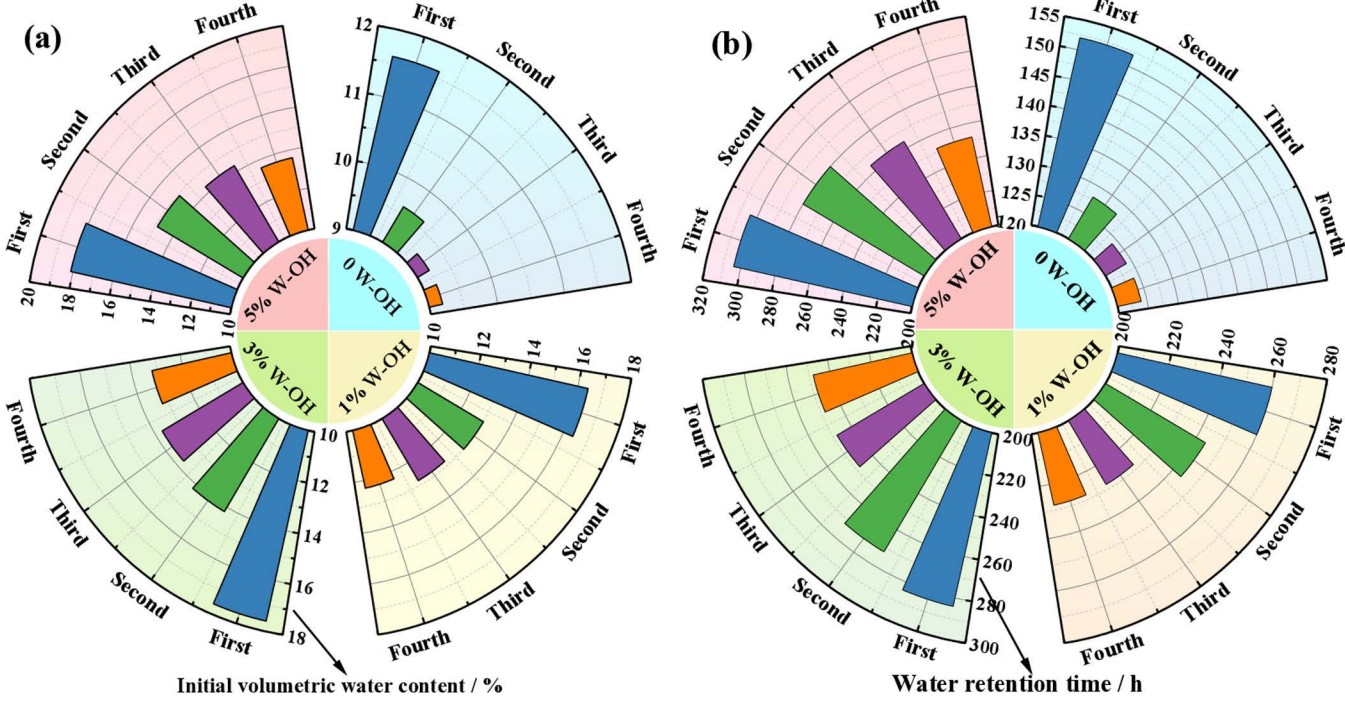

(a) The initial volumetric water content of the samples (b) The drying duration of the samples

**Fig 10. The water retention of the D-W cycles for the samples.** (a) The initial volumetric water content of the samples (b) The drying duration of the samples.

the increase in the concentration of the W-OH solution, for the same D-W cycle. However, with the increase in the D-W cycles, the initial volumetric water content of the samples gradually decreases. The initial volumetric water content of the samples without W-OH gel for the first D-W cycle is 11.6%, and the samples with 1%, 3%, and 5% W-OH solutions are 16.4%, 17.6%, and 18.1%, respectively. The initial volumetric water content of the samples without W-OH gel for the fourth D-W cycle is decreased to 9.0%, and the samples with 1%, 3%, and 5% W-OH solution are decreased to 12.6%, 13.5%, and 14.1%, respectively. That means the W-OH gel gradually deteriorates during the D-W cycles.

Fig 10b shows the drying duration of the samples during the 4 D-W cycles. It is similar to the initial volumetric water content of the samples. The drying duration of the samples gradually decreases with the increase in the D-W cycles. The drying duration of the samples with 1%, 3%, and 5% W-OH solution is decreased to 228 hours, 252 hours, and 260 hours, respectively, for the fourth D-W cycle. They are 87.7%, 87.5%, and 85.5% for the first D-W cycle.

Fig 11 shows the average drying rate of the samples for the D-W cycles. The sample without W-OH gel has the highest average drying rates for every D-W cycle, the average drying rates of the samples with 3% and 5% W-OH solution are close. After the third D-W cycle, the drying rate of each sample reached a stable value.

It can also be seen in Fig 10 that the water retention of all samples decreases gradually and tends to stabilize with the increase in the number of D-W cycles, which means the W-OH gel still maintains better water retention during multiple D-W cycles at the environmental temperature of 30°C.

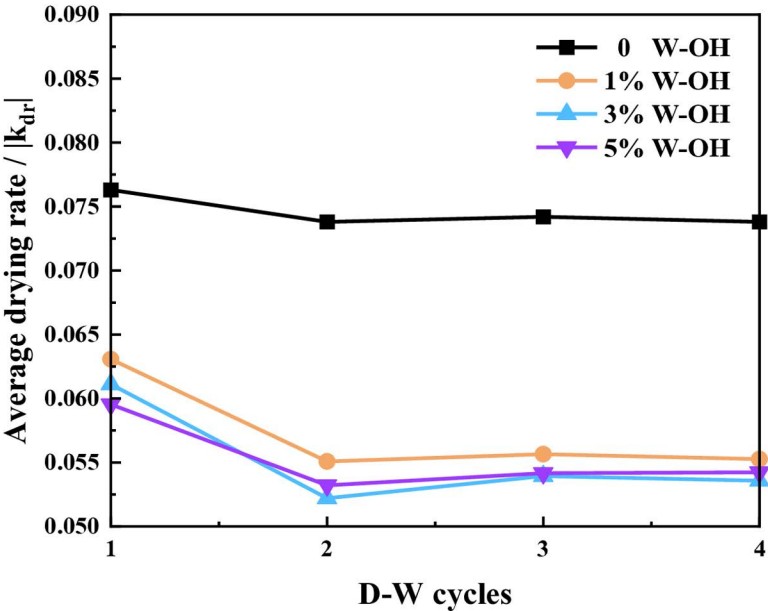

**Fig 11. The average drying rate of the samples for different D-W cycles.**

### Effect of different temperature on the water retention of the waste residue

Fig 12 shows the water retention of different environmental temperatures for the samples with various concentrations of W-OH solution. Due to similar results, only the influence of different temperatures on the water retention of the samples for the 1st and 4th cycles is shown. The initial volumetric water content of the samples for the first D-W cycle is closed when

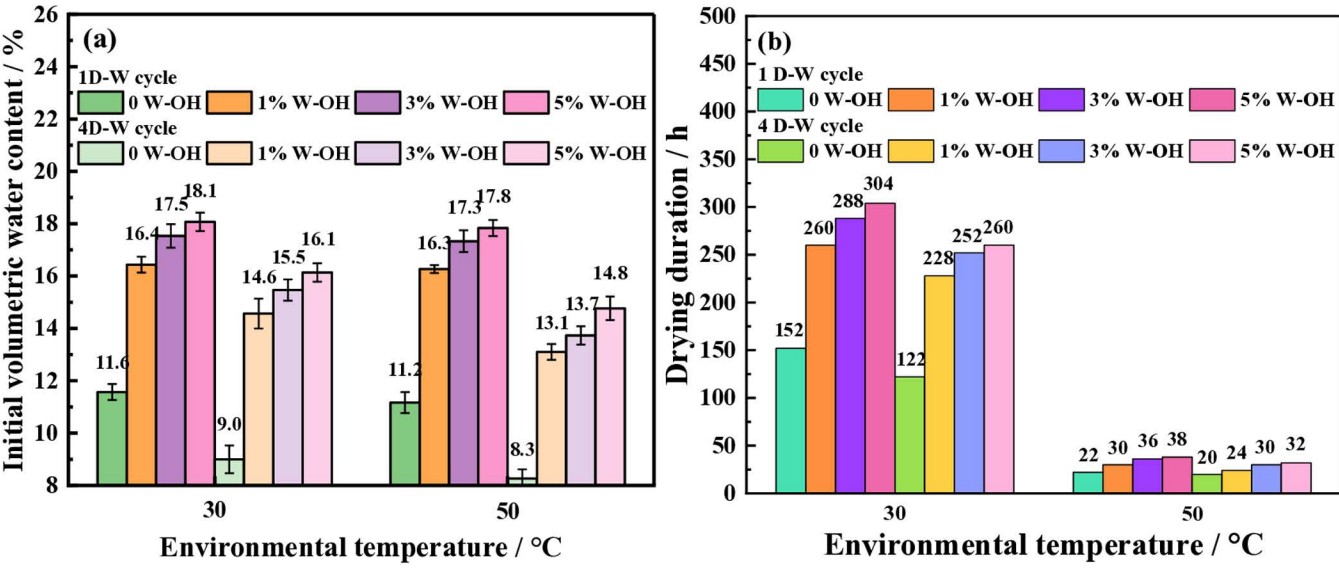

**Fig 12. The water retention of different environmental temperatures for the samples.** (a) The initial volumetric water content of the samples (b) The drying duration of the samples.

the environmental temperatures are 30°C and 50°C. However, for the fourth D-W cycle, the initial volumetric water content of the samples with the same concentration of the W-OH solution at 30°C is higher than at 50°C.

For the samples without W-OH gel, the initial volumetric water content of the samples decreases from 11.6% to 9.0% at 30°C, and from 11.3% to 8.3% at 50°C. The initial volumetric water content of the samples at 30°C and 50°C is closed.

For the samples with a 3% concentration W-OH solution, the initial volumetric water content of the samples for the fourth D-W cycle is from 17.6% to 15.5% at 30°C, but it is from 17.4% to 13.8% at 50°C, and the corresponding decrease percentage is 25% to 34.5%. It can be seen that high temperature has a significant impact on the initial volumetric water content of the particles with W-OH gel. For the samples with W-OH gel, high temperature not only caused the evaporation of water between particles but also resulted in deterioration of the membrane structure. So that the water in the membrane structure was reduced.

Fig 12b shows the drying duration of the samples. For the samples without W-OH gel, the drying duration of the samples is from 152 hours at 30°C to 22 hours at 50°C, and the corresponding decreased percentage is 85.5% for the first D-W cycle. For the samples with 3% W-OH solution, the drying duration of the samples is from 288 hours at 30°C to 36 hours at 50°C, and the corresponding decreased percentage was 87.5%. This demonstrated that the water retention of samples would be greatly weakened under high-temperature conditions, even if the W-OH solution was added.

Fig 13 shows the average drying rate of the samples at 30°C and 50°C. The average drying rate is far higher at 50°C than that at 30°C. In the first D-W cycle and for different concentrations of W-OH solution, it is 0.47 to 0.51 at 50°C, but is 0.074 to 0.076 at 30°C. In the fourth D-W cycle for different concentrations of W-OH solution, it is 0.43 to 0.52 at 50°C, but is 0.053 to 0.059 at 30°C.

The drying resistance of the samples increased with the increase in the concentration of the W-OH solution when the environmental temperature was 30°C. However, the drying resistance of the samples was not good when the samples were subjected to D-W cycles under a high temperature of 50°C.

The deterioration impact of temperature on the water retention of the samples was higher than that of the D-W cycles. That means the high temperature had a greater impact on the structure of the W-OH gel. The coupling effect of the high temperature and D-W cycles resulted in a decrease in the drying resistance of the samples. For the same D-W cycle, the decrease of the average drying rate was close to 30°C with the increase in the concentration of the W-OH solution, but it was gradually decreased with the concentration of the increase in the concentration of the W-OH solution.

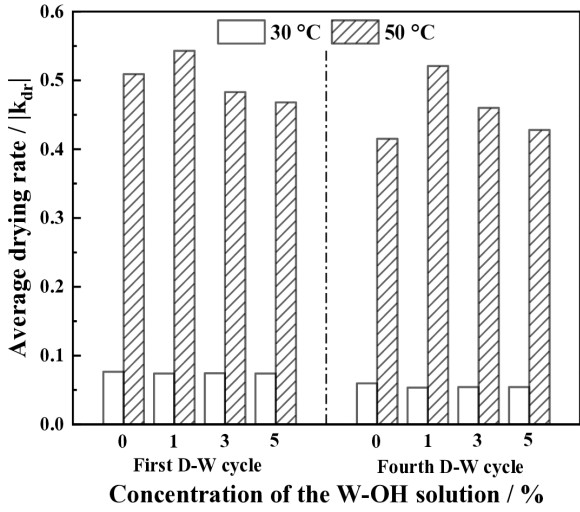

**Fig 13. The average drying rate of the samples at 30°C and 50°C.**

## Discussion

### Mechanism of water retention improvement for the residue treated by W-OH

After the W-OH reacted with water and was added to the residue, a porous gel layer would form (Fig 14). The W-OH gel can attach to the surface of particles and connect multiple particles, and the water existing between particles can not only be affected by the capillary force between particles but also by the encapsulation effect of the W-OH gel. As the concentration of the W-OH solution increased from 1% to 3%, the local membrane structure formed among particles would be thickened. On the one hand, the W-OH gel could narrow or even isolate the water transport channel, making it more difficult for water among particles to be transported to the evaporation surface [27,30,32,38–40]. On the other hand, the isocyanate groups (-NCO) could bond with the water in the pores among particles, and return to the bound water of the W-OH gel, which resulted in the inhibition of evaporation [27,29].

However, the water retention of the samples was not continuously improved as the concentration of the W-OH solution increased from 3% to 5%. This may be because the W-OH solution with a concentration of 3% had already caused a relatively dense structure among the particles. The gel formed by 5% W-OH solution among the particles is locally denser than the 3% W-OH solution, but it also occupies some of the pores between the particles, there is no more volume of pores among the particles to store the water.

### The deterioration mechanism of water retention of the W-OH gel

The water retention capacity of the samples with W-OH gel gradually decreased during 4 D-W cycles, this may be caused by the membrane structure gradually deteriorating [27,40]. As the number of D-W cycles increases, the micro-cracks in the membrane structure may gradually develop. The water retention of the samples with W-OH gel gradually decreased, because of the ruptured and deformed membrane structure, as seen in Fig 15.

Electron microscopic observation of the residue with W-OH gel was carried out using a Hitachi High-tech TM4000Plus scanning electron microscope made in Japan. Fig 16 illustrates the deterioration mechanism of the membrane structure of 3% W-OH gel for temperatures of 30°C and 50°C at the end of the 1st and 4th D-W cycles. Fig 16a shows the membrane

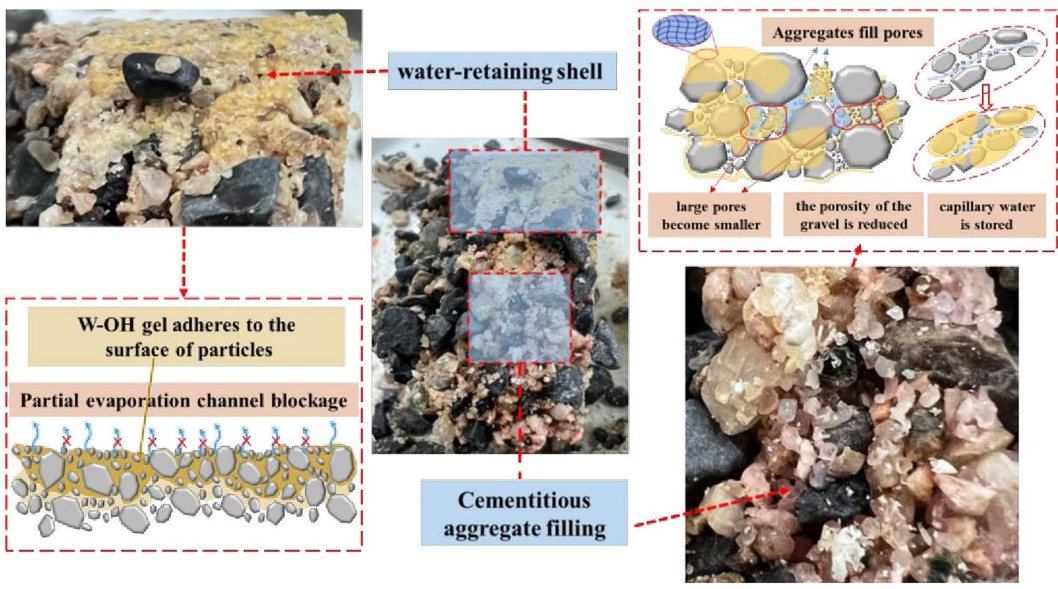

**Fig 14. Microscopic images of W-OH cementitious residue particles.**

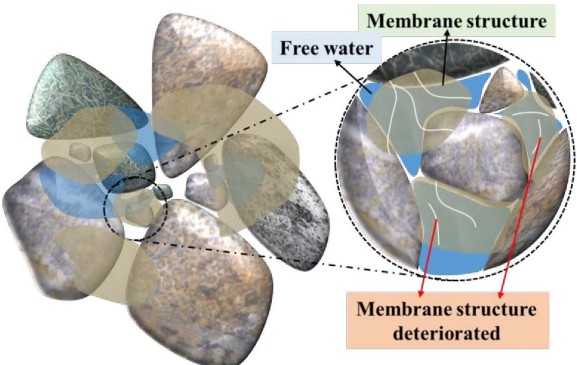

**Fig 15. The membrane structure gradually deteriorates during the D-W cycles.**

structure of W-OH after the first D-W cycles at 30°C. W-OH gel connects the particles, and the intact membrane structure covers the surface of the particles (Fig 16b). After the fourth D-W cycle, some small holes appear in the membrane structure of the W-OH gel (Fig 16c). However, the samples still have good water retention, since the W-OH gel can absorb a lot of water (Fig 16d).

With the increase in the environmental temperature, some large holes appear on the gel membrane among the particles (Fig 16e). Because of the high temperature, the membrane structure becomes thinner, and less water can be held in the W-OH gel (Fig 16f). After the fourth D-W cycle, larger holes appear even when the gel membrane is broken (Fig 16g and 16h). So the water retained in the membrane structure will be lost, and the water retention of the sample will decrease significantly. In addition, the elasticity of the membrane structure of the W-OH gel may be reduced, and the water retention and water absorption capacity will decrease. All of these issues will lead to a decrease in the water retention of the W-OH gel.

## Interactive relationship among the factors

Correlation heatmaps reflect the correlation between various variables and are widely used in courses such as medicine, agriculture and forestry, architecture, computer science, and environmental science. The Pearson correlation coefficient is applicable to linear correlation between variables (red indicates positive correlation, blue indicates negative correlation, the smaller the P-value, the more significant the correlation). Fig 17 shows the relationship between W-OH concentration, D-W cycle times, ambient temperature, sample volume water content, and water retention time. There is a significant positive correlation between volumetric water content and solution concentration, a significant negative correlation between volumetric water content and D-W cycle, and a significant negative correlation between water retention time and environmental temperature.

## Plant growth test

Fig 18 shows the growth state of Amorpha fruticosa for the observation for 21 days. The residue without W-OH (see Fig 18a) was worse than the residue with 3% W-OH (see Fig 18b).

Fig 18c shows the root growth state of Amorpha fruticosa after 21d, compared to the residue without W-OH. The average root length and the average plant height of Amorpha fruticosa in residue with 3% W-OH had a better growth state. The top cotyledons were more stretchy, the stems were more robust, and root elongation and branching. Fig 18d shows the fixation effect of roots on fine particles of Amorpha fruticosa, and the W-OH binds the fine particles and wraps them around the roots of Amorpha fruticosa.

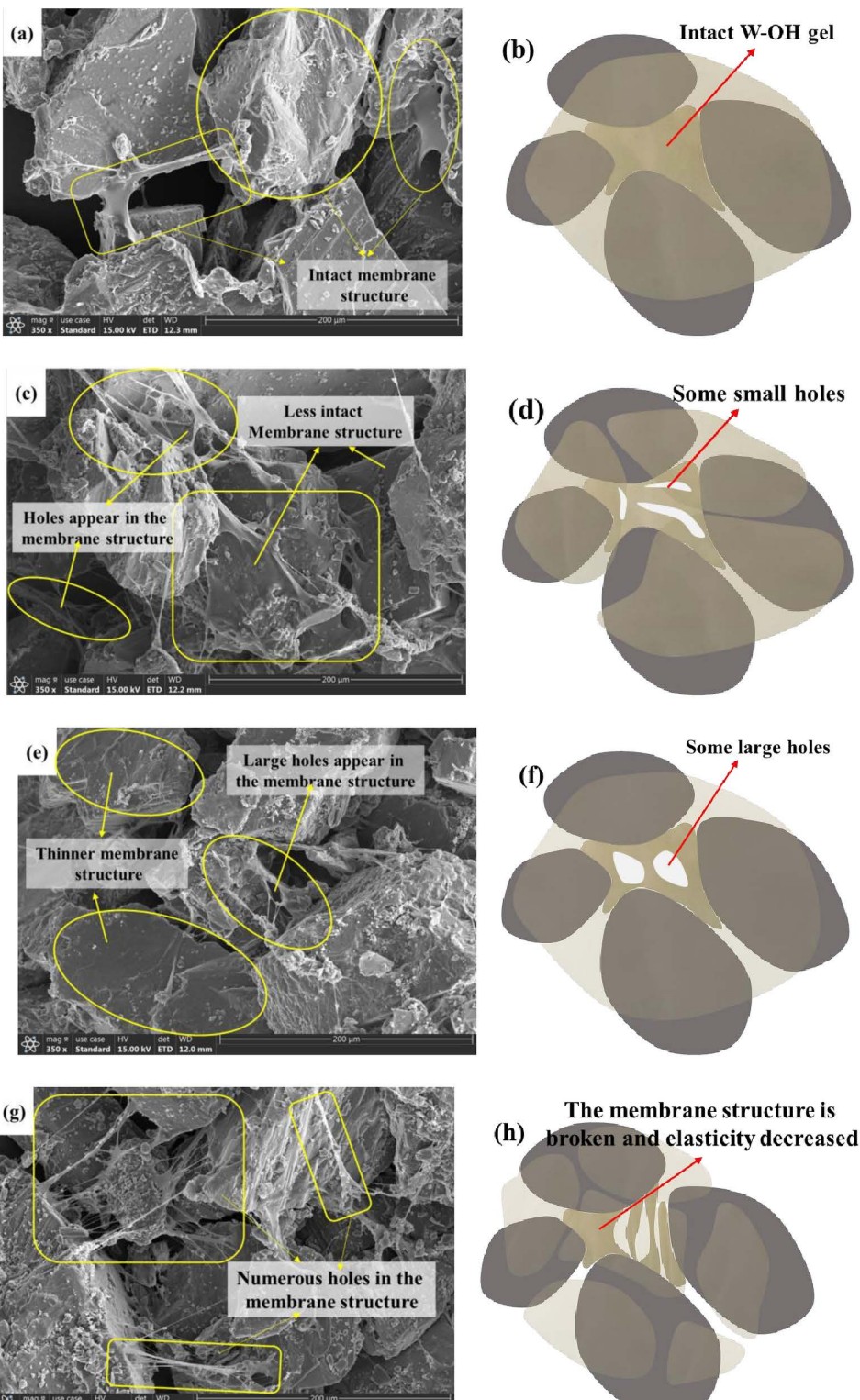

**Fig 16. The deterioration of residue with 3% W-OH for different temperatures and D-W cycles.** (a) and (c) The membrane structure of W-OH at 30°C of 1st and 4th D-W cycles (Zoom in 350 times) (e) and (g) The membrane structure of W-OH at 50°C of 1st and 4th D-W cycles (Zoom in 350 times). (b), (d), (f), (h) The states of the samples.

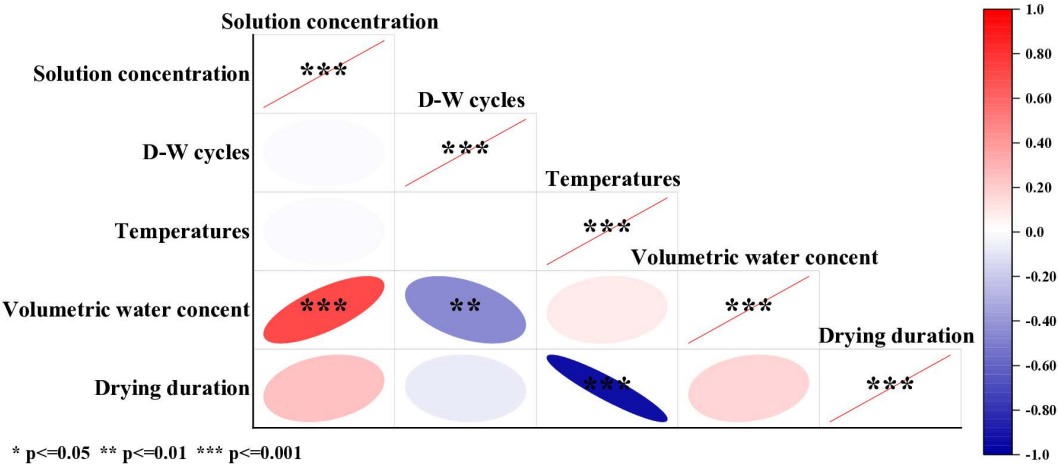

**Fig 17.  Interactive relationship among the factors.**

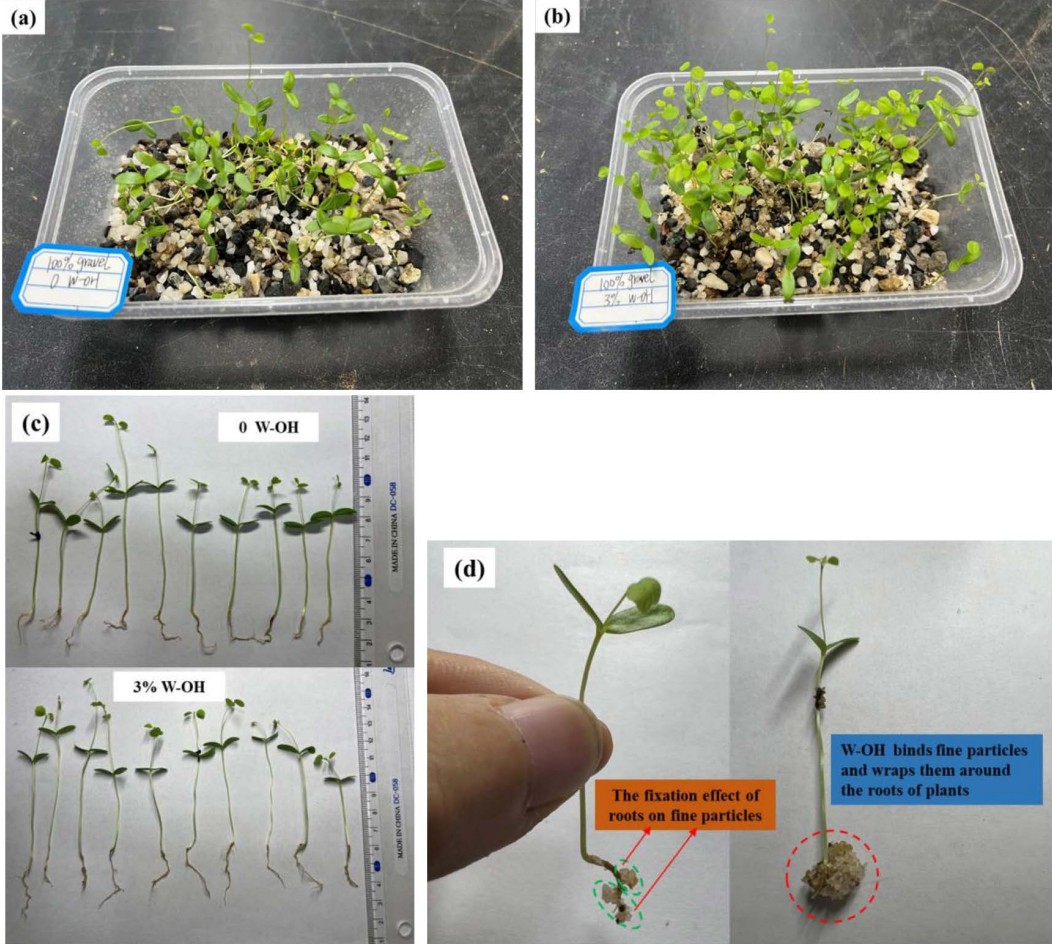

**Fig 18.  The growth state of amorpha fruticosa for the observation for 21 days.**

Fig 19 Statistics the percentage of germination, the average plant height, the mean root length, the root flesh weight, and the root-shoot ratio of the Amorpha fruticosa in residues. The various indices of Amorpha fruticosa in residue with 3% W-OH were better than residue without W-OH. The percentage of germination of the waste gravel with 3% W-OH was 94% but the residue without W-OH was 85%. The average plant height of the waste gravel with 3% W-OH was 73 mm, but the residue without W-OH was 66 mm. The mean root length (mm), root flesh weight (mg), and root-shoot ratio (%) of the waste gravel were 1.60, 2.0, and 1.82 times those of the waste gravel without W-OH, respectively.

## Conclusions

To improve the water retention of waste residue for vegetation restoration, the effects of the residues adding modified hydrophilic polyurethane (W-OH) were investigated. The influence of the concentration of W-OH solution, the number of D-W cycles, and the temperature on the water retention of the waste residue before and after improvement was studied. Some conclusions can be drawn as follows.

[1] The water retention of the waste residue treated by W-OH is significantly affected by the concentration of the W-OH solution and the number of D-W cycles. The initial volumetric water content of the sample increased, and the drying duration was extended as the concentration of the W-OH solution increased. The initial volumetric water content was decreased, and the drying duration was shortened with the increase in D-W cycles.

[2] The higher temperature has a significant impact on the water retention of the residue treated by the W-OH gel. For the samples with the same concentration of W-OH solution, the initial volumetric water content of the samples decreased, and the drying duration reduced as the environmental temperature went from 30°C to 50°C.

[3] Adding W-OH gel can effectively increase the water retention of the waste residue because it can reduce or even block the pores among the residue particles. However, the stability of the membrane structure of the W-OH gel is affected by temperature and the number of cycles. After multiple D-W cycles, the holes appear in the membrane structure. As the temperature increases, the membrane structure becomes thin and elastic, and the water absorption

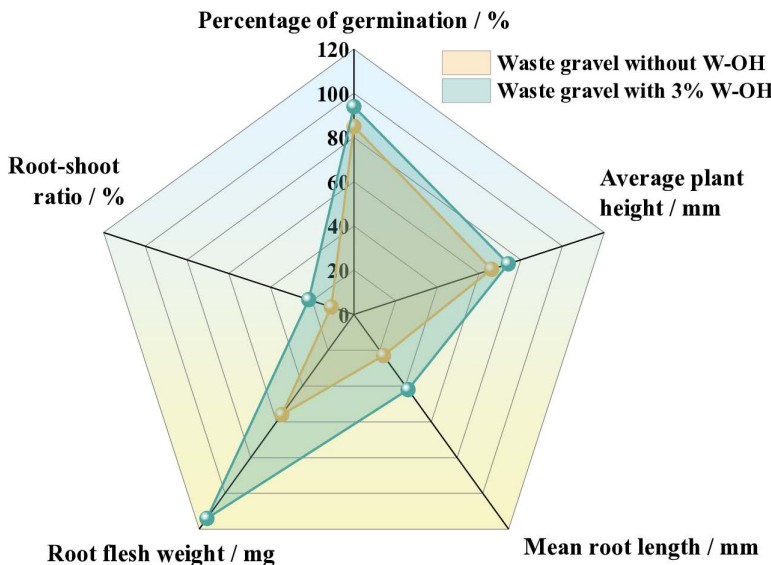

**Fig 19. The growth indexes of amorpha fruticosa for the observation for 21 days.**

is gradually reduced. After multiple D-W cycles, larger holes or even fractures may appear in the membrane structure of the W-OH gel.

[4] The plant growth test shows that the 3% W-OH provides sufficient water for the growth of Amorpha fruticosa in the residue. The roots of Amorpha fruticosa fixed the fine residue particles, and the W-OH binds the fine residue particles and wraps them around the roots of the plant. Various growth indicators of Amorpha fruticosa in residues with 3% W-OH were better than the residue without W-OH.

It is worth noting that this article only analyzed the effect of W-OH material on the water retention and vegetation potential of residue at the experimental level. For the degradation process of W-OH material, Fourier transform infrared spectroscopy (FTIR) and X-ray diffraction (XRD) tests will be conducted later to quantitatively analyze its degradation effect under temperature and ultraviolet radiation. In order to study the improvement of water retention performance of residue by W-OH, waste residue has been scaled down, but there is still no field test verification. Therefore, the field test and field-scale grass planting experiment will be carried out in the future. In the later stage, it will conduct more in-depth experiments to explore the feasibility of applying W-OH in waste disposal sites from multiple perspectives.

## Supporting information

**S1 File. Figures**
(ZIP)

**S2 File. Tables**
(DOC)

## Author contributions

**Conceptualization:** Jun Kang Zhao, Qun Chen, Ting Quan He.

**Data curation:** Lu Li.

**Funding acquisition:** Qun Chen.

**Investigation:** Lu Li.

**Methodology:** Jun Kang Zhao.

**Project administration:** Qun Chen.

**Supervision:** Jun Kang Zhao.

**Validation:** Lu Li.

**Writing – original draft:** Lu Li.

**Writing – review & editing:** Jun Kang Zhao, Qun Chen, Cheng Zhou.

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
