## [Decision Letter · Decision Letter 0]

12 Aug 2025

Dear Dr. chen,

Thank you for submitting your manuscript to PLOS ONE. After careful consideration, we feel that it has merit but does not fully meet PLOS ONE’s publication criteria as it currently stands. Therefore, we invite you to submit a revised version of the manuscript that addresses the points raised during the review process.

We look forward to receiving your revised manuscript.

Kind regards,

Muammar Qadafi

Academic Editor

PLOS ONE

Journal Requirements:

“This research was substantially supported by the Sichuan Province National Natural Science Foundation of China (Grant No. 24NSFSC0260); and the 3rd series of the Guangxi Transportation Keynote Research Program in 2020 (Grant No. XFZ-KY-LYH-20200112).“

“This research was substantially supported by the Sichuan Province National Natural Science Foundation of China (Grant No. 24NSFSC0260); and the 3rd series of the Guangxi Transportation Keynote Research Program in 2020 (Grant No. XFZ-KY-LYH-20200112). The authors are grateful to the anonymous reviewers for their valuable comments and suggestions that helped improve the quality of this manuscript.”

“This research was substantially supported by the Sichuan Province National Natural Science Foundation of China (Grant No. 24NSFSC0260); and the 3rd series of the Guangxi Transportation Keynote Research Program in 2020 (Grant No. XFZ-KY-LYH-20200112).“

5. We note that your Data Availability Statement is currently as follows: All relevant data are within the manuscript and its Supporting Information files

6. PLOS requires an ORCID iD for the corresponding author in Editorial Manager on papers submitted after December 6th, 2016. Please ensure that you have an ORCID iD and that it is validated in Editorial Manager. To do this, go to ‘Update my Information’ (in the upper left-hand corner of the main menu), and click on the Fetch/Validate link next to the ORCID field. This will take you to the ORCID site and allow you to create a new iD or authenticate a pre-existing iD in Editorial Manager.

Reviewers' comments:

Reviewer's Responses to Questions

**Comments to the Author**

1. Is the manuscript technically sound, and do the data support the conclusions?

Reviewer #1: Yes

Reviewer #2: Yes

2. Has the statistical analysis been performed appropriately and rigorously?

Reviewer #1: No

Reviewer #2: No

3. Have the authors made all data underlying the findings in their manuscript fully available?

Reviewer #1: Yes

Reviewer #2: Yes

4. Is the manuscript presented in an intelligible fashion and written in standard English?

Reviewer #1: Yes

Reviewer #2: Yes

Reviewer #1: Abstract: should be precise and add some readings containing positive outcomes. Conclusion is not drawn.

Introduction: Self explanatory and well written.

Methods: Methodology is not well explained. Mostly results mentioned in this section. Please revise this section with appropriate procedures. Figures should be merged to reduce numbers. No statistical data mentioned.

Results and discussion: Well explained but some part mentioned in methodology. So, rewrite the results.

Conclusion: well written and precise. Acknowledgment should be written as Acknowledgements.

Abbreviations should be self explanatory throughout the manuscript.

Reviewer #2: Comment

The sentence in line 29-31 make the experiment void because you were going to amend the waste residue or soil is arid area natural this area is known by high temperature and erratic rainfall which weaken water retention of the waste residue. So why you are wasting your resources or where did you were recommend such treatments?

The abstract lack specific conclusion and implication of your findings.

31 gradually deteriorated, so that the water retention of the waste residue weakened

Line 58 and 59 were not consistent with so either remove add in appropriate paragraph.

Did you sow the test plant on that soil? If so please explain detail procedures.

If they grow voluntarily show us the reasons behind and either any other plants grow on it.

No any statistically tools were explained to analysis data or test results

**Do you want your identity to be public for this peer review?** For information about this choice, including consent withdrawal, please see our Privacy Policy

Reviewer #1: **Yes: ** Saiqa Andleeb

Reviewer #2: No

---

## [Author Response · Author response to Decision Letter 1]

26 Aug 2025

Author Responses:

Reviewer #1:

Comment 1: " Abstract: should be precise and add some readings containing positive outcomes. Conclusion is not drawn. "

Response: Thank you very much for your comment! Regarding the lack of data display in the abstract section that you mentioned, we have refined and supplemented the data in the abstract section. Please refer to the red font in the text for specific modifications.

Modified content:

Abstract The large pores and lack of water storage capacity limit the ecological rehabilitation of the waste residue. Modified hydrophilic polyurethane (W-OH) was used to improve the water retention of the residue. The water retention capacity of residues with different mass concentrations (1%, 3%, and 5%) of W-OH solution based on water mass was evaluated for several drying-wetting (D-W) cycles at 30°C and 50°C. The plant growth experiment investigated the plant growth status of waste residue before and after adding W-OH, to demonstrate the excellent water retention of W-OH under the same waste residue. Infiltration tests showed that W-OH effectively increased the volumetric water content (VWC) of waste residue and reduced its drying time. Untreated residue had 11.62% saturated initial VWC and 152 h drying time; 3% and 5% W-OH groups showed similar results (17.63%-18.09% VWC, 288-304 h drying time), so 3% W-OH is recommended economically. After 4 dry-wet cycles, both groups had reduced VWC and drying time. However, the 3% W-OH-treated residue still exhibited better water retention. A dense membrane is formed by W-OH solution around waste residue particles, enclosing/connecting particles, filling pores (reducing large ones), increasing capillary water storage, and hindering water evaporation. Degraded by more D-W cycles and higher temperature, the membrane weakens water retention, yet W-OH-treated residue still has better water retention than the untreated residue. Water retention of improved waste residue was enhanced, and plant growth promoted, per plant growth tests. After 21-day planting, Amorpha fruticosa in 3% W-OH-treated residue had better growth parameters than in untreated residue.

Comment 2: " Introduction: Self explanatory and well written. "

Response: Thank you very much for your comment. Thank you very much for your recognition of this section!

Comment 3: " Methods: Methodology is not well explained. Mostly results mentioned in this section. Please revise this section with appropriate procedures. Figures should be merged to reduce numbers. No statistical data mentioned. "

Response: Thank you very much for your comment! In response to the issues you mentioned regarding the experimental methods, we have provided additional explanations on the details of the experimental methods, which can enhance the reproducibility of the experiment. Please refer to the red font in the text for specific modifications.

Modified content:

Test methods

Dry-wet cycle test at different temperatures

To investigate the water retention durability of the W-OH gel, and the effect of environmental temperature. The saturated residue samples containing W-OH were placed in a drying oven at 30 °C and 50°C, and the mass of the samples was weighed every 2 hours to calculate the volumetric water content of the samples. Once the volumetric water content of the samples dropped to zero, which means once of the D-W cycles was over. Deionized water is reapplied to the sample, until the amount of water seeping out from the bottom of the sample is equal to the amount of water dripping in, at which point the sample reaches saturation again. Then the subsequent wet-dry cycle test is initiated. This wet-dry cycling procedure is repeated four times to evaluate the water retention durability of W-OH-treated sand-residue composites. In the drying duration, the slope of the tangent between the initial volumetric water content of the sample and the drying duration is used to define the drying rate | kdr | (% / h) of the sample after different concentrations of improvement (eq.1).

(1)

Plant growth test

A plant growth test at laboratory was conducted after the best concentration of the W-OH solution was determined. Amorpha fruticosa was chosen as the herbaceous plant for the experiment. Amorpha fruticosa grass seeds were sown in transparent plastic sample boxes with perforations at the bottom. The shape of the sample box is trapezoidal, with a height of 600mm. The top length and width of the sample box are 170mm and 117mm respectively, the bottom length and width are 135mm and 85mm respectively, and the length and width of the box at a height of 40mm are 156mm and 106mm respectively. Based on the density of the waste slag, the required mass to fill 40mm is calculated, and the waste slag is loaded into the sample box. Evenly sprinkle 100 full particle Amorpha fruticosa grass seeds into the sample box containing waste residue, and then drip 2L/m2 of water and 3% concentration of W-OH solution into the sample box.

The growth of plants in waste residue at a concentration of 3% and without W-OH for 21 days were studied. During the growth process of the plant, drip 2L/m2 of water into the sample box every 7 days. The condition of the plant roots was recorded and the mass of both plant roots and stems was weighed using an analytical balance. The root top ratio [r/t] (Root biomass (dry weight) / Aboveground biomass (dry weight)) was then calculated. It is worth noting that in the plant growth experiment, 10 plants were selected for the determination of growth parameters, and the average value was taken as the basis for judging the growth effect of plants in the waste Residue.

Comment 4: "Results and discussion: Well explained but some part mentioned in methodology. So, rewrite the results. "

Response: Thank you very much for your comment! We will make certain modifications to the results and discussion sections of the article.

Comment 5: " Conclusion: well written and precise. Acknowledgment should be written as Acknowledgements. Abbreviations should be self explanatory throughout the manuscript."

Response: Thank you very much for your comment! Thank you very much for recognizing our research content.

Thank you again for reading our paper and providing constructive feedback, which has helped to make the structure and content of this paper more rigorous!

Reviewer #2:

Comment 1: " The sentence in line 29-31 make the experiment void because you were going to amend the waste residue or soil is arid area natural this area is known by high temperature and erratic rainfall which weaken water retention of the waste residue. So why you are wasting your resources or where did you were recommend such treatments?"

Response: Thank you very much for your comment. Due to the large amount of waste generated during excavation and blasting in mines and mountainous areas, which cannot be utilized in a timely manner, it has adverse effects on the surrounding ecological environment. Therefore, it is necessary to carry out long-term vegetation restoration to facilitate environmentally friendly disposal before finding suitable reuse methods in the later stage. It is necessary to find an efficient water retention solution for the broken waste and weak water retention. This experiment aims to improve the water retention performance of waste slag by applying the environmentally friendly water retention and sand fixation material W-OH to the practice of improving the water retention performance of waste slag. After selecting the appropriate W-OH concentration, preliminary simple indoor plant experiments were conducted to verify the good water retention performance of W-OH. Chemical materials deteriorate under the influence of high temperature and wet dry cycles, so we conducted experiments to explore. The results showed that although W-OH materials are weakened by high temperature and wet dry cycles, their water retention is still much better than untreated waste.

Comment 2: " The abstract lack specific conclusion and implication of your findings."

Response: Thank you for your comment! We have supplemented and refined the abstract section. Please refer to the red font in the text for specific modifications.

Modified content:

Abstract The large pores and lack of water storage capacity limit the ecological rehabilitation of the waste residue. Modified hydrophilic polyurethane (W-OH) was used to improve the water retention of the residue. The water retention capacity of residues with different mass concentrations (1%, 3%, and 5%) of W-OH solution based on water mass was evaluated for several drying-wetting (D-W) cycles at 30°C and 50°C. The plant growth experiment investigated the plant growth status of waste residue before and after adding W-OH, to demonstrate the excellent water retention of W-OH under the same waste residue. Infiltration tests showed that W-OH effectively increased the volumetric water content (VWC) of waste residue and reduced its drying time. Untreated residue had 11.62% saturated initial VWC and 152 h drying time; 3% and 5% W-OH groups showed similar results (17.63%-18.09% VWC, 288-304 h drying time), so 3% W-OH is recommended economically. After 4 dry-wet cycles, both groups had reduced VWC and drying time. However, the 3% W-OH-treated residue still exhibited better water retention. A dense membrane is formed by W-OH solution around waste residue particles, enclosing/connecting particles, filling pores (reducing large ones), increasing capillary water storage, and hindering water evaporation. Degraded by more D-W cycles and higher temperature, the membrane weakens water retention, yet W-OH-treated residue still has better water retention than the untreated residue. Water retention of improved waste residue was enhanced, and plant growth promoted, per plant growth tests. After 21-day planting, Amorpha fruticosa in 3% W-OH-treated residue had better growth parameters than in untreated residue.

Comment 3: "31 gradually deteriorated, so that the water retention of the waste residue weakened. Line 58 and 59 were not consistent with so either remove add in appropriate paragraph.."

Response: Thank you for your comment! Your comment is correct! Due to the low relevance of these two sentences to the research content, we have deleted this part of the content.

Comment 4: " Did you sow the test plant on that soil? If so please explain detail procedures."

Response: Thank you for your comment. In this paper, we only conducted indoor vegetation experiments on waste samples with W-OH added, using plastic sample box filling experiments. The plant species was amorpha fruticosa, and 100 full particle amorpha fruticosa grass seeds were tested for 21 days. The experimental results showed that the growth effect of amorpha fruticosa plants in the waste treated with W-OH was better than that of untreated waste samples. In the later stage of the experiment, we will compare and screen multiple plants, as well as conduct on-site experiments (see Fig. 18a and 18b). Please refer to the red font in the text for specific modifications.

Modified content:

Plant growth test

A plant growth test at laboratory was conducted after the best concentration of the W-OH solution was determined. Amorpha fruticosa was chosen as the herbaceous plant for the experiment. Amorpha fruticosa grass seeds were sown in transparent plastic sample boxes with perforations at the bottom. The shape of the sample box is trapezoidal, with a height of 600mm. The top length and width of the sample box are 170mm and 117mm respectively, the bottom length and width are 135mm and 85mm respectively, and the length and width of the box at a height of 40mm are 156mm and 106mm respectively. Based on the density of the waste slag, the required mass to fill 40mm is calculated, and the waste slag is loaded into the sample box. Evenly sprinkle 100 full particle Amorpha fruticosa grass seeds into the sample box containing waste residue, and then drip 2L/m2 of water and 3% concentration of W-OH solution into the sample box.

The growth of plants in waste residue at a concentration of 3% and without W-OH for 21 days were studied. During the growth process of the plant, drip 2L/m2 of water into the sample box every 7 days. The condition of the plant roots was recorded and the mass of both plant roots and stems was weighed using an analytical balance. The root top ratio [r/t] (Root biomass (dry weight) / Aboveground biomass (dry weight)) was then calculated. It is worth noting that in the plant growth experiment, 10 plants were selected for the determination of growth parameters, and the average value was taken as the basis for judging the growth effect of plants in the waste Residue.

Comment 5: " If they grow voluntarily show us the reasons behind and either any other plants grow on it."

Response: Thank you for your comment. The plant selected in this experiment is amorpha fruticosa, which has good drought resistance and environmental adaptability. The sample is watered every 7 days, and the indoor experiment is minimally affected by the natural environment. Therefore, amorpha fruticosa plants can also grow normally in untreated gravel. However, in comparison, amorpha fruticosa grown in the waste residue after W-OH treatment showed better growth, with better growth parameters such as germination rate, root length, plant height, root fresh weight, and root shoot ratio.

Comment 6: " No any statistically tools were explained to analysis data or test results."

Response: Thank you for your comment! This experiment mainly focused on the improvement effect of different concentrations of W-OH on the water retention of waste slag, with few influencing factors involved, and did not consider the influence of complex environments. The experimental data is centralized and single, but it can still well illustrate the water retention of W-OH and the effects of temperature and wet dry cycles on the water retention of W-OH materials. In the later stage, we will conduct more in-depth experiments to explore the feasibility of applying W-OH in waste disposal sites from multiple perspectives.

Thank you again for reading our paper and providing constructive feedback, which has helped to make the structure and content of this paper more rigorous!

Editor:

Comment 1: "Please ensure that your manuscript meets PLOS ONE's style requirements, including those for file naming. "

Response: Dear editorial teacher! In this revised manuscript, we have adjusted the text titles, tables, graphics, and other formats according to the requirements of the journal. Please refer to the modified content in red in the text.

Comment 2: "In your Methods section, please provide additional information regarding the permits you obtained for the work. Please ensure you have included the full name of the authority that approved the field site access and, if no permits were required, a brief statement explaining why."

Response: Dear editorial teacher! The experimental methods in this article are all aimed at conducting indoor experiments, as an exploratory study for the promotion and application of W-OH materials in the later stage. And there have been many studies using W-OH in engineering applications such as soil stabilization and crack resistance, achieving rich results and good progress.

Comment 3: " Thank you for stating the following financial disclosure:

“This research was substantially supported by the Sichuan Province National Natural Science Foundation of China (Grant No. 24NSFSC0260); and the 3rd series of the Guangxi Transportation Keynote Research Program in 2020 (Grant No. XFZ-KY-LYH-20200112).“

Please include this amended Role of Funder statement in your cover letter; we will change the online submission form on your behalf"

Response: Dear editorial teacher! We have added information about the author's contribution in the

---

## [Decision Letter · Decision Letter 1]

1 Sep 2025

Environmentally friendly polymers are used to enhance the water retention capacity of waste residue and the potential for vegetation growth

PONE-D-25-39615R1

Dear Dr. Qun chen,

We’re pleased to inform you that your manuscript has been judged scientifically suitable for publication and will be formally accepted for publication once it meets all outstanding technical requirements.

Kind regards,

Muammar Qadafi

Academic Editor

PLOS ONE

Additional Editor Comments (optional):

Reviewer #1:

Reviewer #2:

Reviewers' comments:

Reviewer's Responses to Questions

**Comments to the Author**

Reviewer #1: All comments have been addressed

Reviewer #2: All comments have been addressed

2. Is the manuscript technically sound, and do the data support the conclusions?

Reviewer #1: Yes

Reviewer #2: Yes

3. Has the statistical analysis been performed appropriately and rigorously?

Reviewer #1: Yes

Reviewer #2: Yes

4. Have the authors made all data underlying the findings in their manuscript fully available?

Reviewer #1: Yes

Reviewer #2: Yes

5. Is the manuscript presented in an intelligible fashion and written in standard English?

Reviewer #1: Yes

Reviewer #2: Yes

Reviewer #1: Authors did research on "Environmentally friendly polymers are used to enhance the water retention capacity of waste residue and the potential for vegetation growth". All corrections amneded and Congratulations to all authors for innovative study.

Reviewer #2: Dear authors thank you for your clarification and amendments accordingly. Yet the only questions you didn't addressed is:

1. In abstract section you said "After 21-day planting, Amorpha fruticosa in 3% W-OH-treated residue had better growth parameters than in untreated residue" so what it implies or what did you deduct from this scientifical for any concerningbody.

2. 》》 you recommended 3% W-OH is recommended economically without conducted any economically analysis procedures so it is better to replace with other appropriate sentences or indicate the econimic value it adds.

In discussion section

Mostly if it is not mandatory ( incase the value have additional implication), referring tables and figures in your results section is not recommended. However you can reason out the possibility of changes or improvements occurring due to your treatments compared to control.

In line 274 Effect of environmental temperature(What does it mean?)

I am not comfortable with such ambiguous words. Because you are control the temperature in your laboratory so it is better to replace with "Effect of different temperature on..(water retention...).

Under plant growth test you only provide the results without any discussion and scientific reasoning and supportive references of related topics or researchers.

##With this I hope you will make it meaningful scientific reasoning with in depth research in future ( ^this in not related to this manuscript but to remind you again ^****)###.

"In the later stage, we will conduct more in-depth experiments

to explore the feasibility of applying W-OH in waste disposal sites from

multiple perspectives".

**Do you want your identity to be public for this peer review?** For information about this choice, including consent withdrawal, please see our Privacy Policy

Reviewer #1: **Yes: ** Saiqa Andleeb

Reviewer #2: No

---

## [Editor Report · Acceptance letter]

PONE-D-25-39615R1

PLOS ONE

Dear Dr. chen,

I'm pleased to inform you that your manuscript has been deemed suitable for publication in PLOS ONE. Congratulations! Your manuscript is now being handed over to our production team.

Kind regards,

on behalf of

Dr. Muammar Qadafi

Academic Editor

PLOS ONE